# Optimal Design of Double-Pole Magnetization BLDC Motor and Comparison with Single-Pole Magnetization BLDC Motor in Terms of Electromagnetic Performance

**Hyo-Seob Shin** [1] , **Gang-Hyeon Jang** [1], **Kyung-Hun Jung** [2], **Seong-Kook Cho** [2], **Jang-Young Choi** [1,*] **and Hyeon-Jae Shin** [2]

1 Department of Electrical Engineering, Chungnam National University, Daejeon 34134, Korea; shs1027@cnu.ac.kr (H.-S.S.); gh.jang@cnu.ac.kr (G.-H.J.)
2 Research and Development Center, Hanon Systems, Daejeon 34325, Korea; kjung1@hanonsystems.com (K.-H.J.); scho1@hanonsystems.com (S.-K.C.); hshin4@hanonsystems.com (H.-J.S.)
* Correspondence: choi_jy@cnu.ac.kr

**Abstract:** This study presents an optimal double-pole magnetization brushless DC (BLDC) motor design, compared to a single-pole magnetization BLDC motor in terms of electromagnetic performance. Initially, a double-pole model is selected based on the permanent magnet (PM) of the single-pole model. The pole separation space, which is generated in the magnetization process of the double-pole PM, is selected based on the pole space of the single-pole model. Moreover, the PM offset is selected considering the PM volume of the single-pole model. Further, an optimal model is selected using the multiple response optimal method, which is a type of response surface methodology (RSM). The objective of the optimal design is to maintain the back EMF and decrease the cogging torque; the design variables include the pole separation space and PM offset. The experimental points of the initial model are designed using the central composite method (CCD). Finally, the optimization is verified by comparing the experimental and analysis results of the single-pole model with the analysis results of the optimal model.

**Keywords:** BLDC motor; double-pole magnetization; RSM

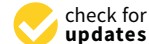



## 1. Introduction

BLDC (brushless DC) motors use electronic rectifying equipment instead of mechanical equipment such as brushes and commutators. Therefore, their maintenance cost is low compared to DC motors. Moreover, because of the low cost driving devices and the development of control technology, BLDC motors have been replacing DC motors in various fields [1–4]. As a double-pole magnetization permanent magnet (PM) has two poles in one magnet, even if the number of magnets is halved, the same number of poles can be obtained. When the number of poles increases in a single-pole magnetization BLDC motor, it becomes difficult to assemble the PMs due to the magnetic forces between adjacent PMs; in addition, they become more unstable at high rotational speeds. Thus, halving the number of PMs using the double-pole PMs not only reduces the assembly time and manufacturing tolerances, but also provides stability at high rotational speeds [5,6].

In the research field of multi-pole magnetization PM, the majority of studies are on the ring magnet. In [7], the construction of a magnetization head for 8-poles magnetization of ring magnets is discussed. In [8], the radial force densities of ring magnet are discussed depending on the magnetization pattern. In [9], the advantages of multi-pole dielectromagnets, allowing the replacement of the multi-pole glued constructions of bipolar permanent magnets, is presented. In [10], an exact analytical model to compute the magnetic field generated by a diametrically magnetized cylindrical/ring shape permanent magnet is

presented. In [11], an easily computable 3D filed solution for axially polarized multi-pole ring and disk magnets is presented. Although there are several studies on multi-pole magnetization ring magnets, it is difficult to find studies on double-pole magnetization PM motors and their optimized magnet shape.

Therefore, this study presents an optimal design for a double-pole magnetization BLDC motor, compared to a single-pole magnetization BLDC motor in terms of the electromagnetic performance. Initially, a double-pole model is selected, which is based on the PM of the single-pole model. All the other structures are identical except for the PMs. Pole separation space is generated in the magnetization process of the double-pole PM due to the structure of the magnetizer. This space includes small flux; it is assumed that this flux decreases linearly in this study. The length of the pole separation space in the initial model is selected based on this assumption. Moreover, as the number of the PM offset parts is halved in the double-pole model, the length of the PM offset is selected considering the PM volume. Electromagnetic analysis results of single-pole and initial double-pole models from finite element methodology (FEM) were compared and confirmed the need for optimization. An optimal double-pole model is selected using the multiple response optimal method, which is a type of response surface methodology (RSM). The objective of optimization is to maintain the back EMF and reduce the cogging torque; the design variables include the pole separation space and PM offset. The experimental points of the initial double-pole model are designed using central composite method (CCD), which is a type of design method for experimental points, and is generally used in RSM. The quadratic regression model can be obtained from the response values of each point, and the optimal design variables can be selected from the regression model [12]. Finally, the optimal design is verified by comparing the experimental and analysis results of the single-pole model with the analysis results of the optimal double-pole model. Additionally, the electromagnetic performance is compared, and the results of the experiment and analysis are discussed.

## 2. Initial Double-Pole Magnetization BLDC Motor

### 2.1. Analysis Model

Figure 1 shows the two analysis models and their flux paths. Winding phases and their direction are marked in the slot of stator core. Figure 1a depicts a single-pole magnetization BLDC motor, with 10 magnets, whereas Figure 1b depicts the initial double-pole magnetization BLDC motor with 5 magnets. The two models are the outer rotor type BLDC motors with ten poles each. All the structures are identical except for the PMs. Dimensions of the analysis model are shown in Table 1. All dimensions except PM embrace are identical in the analysis models. Moreover, PM specifications are shown in Table 2.

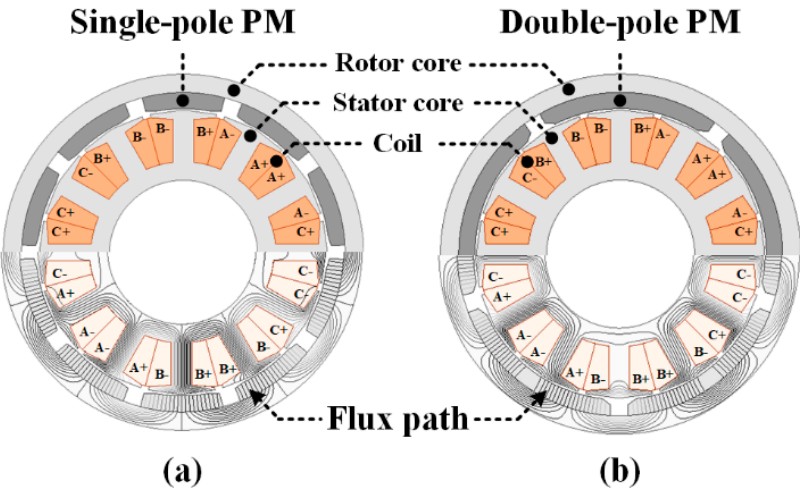

**Figure 1.** Analysis models and their flux path: (**a**) Single-pole magnetization BLDC (brushless DC) motor and (**b**) initial double-pole magnetization BLDC motor.

**Table 1.** Dimensions of the analysis motor.

| Item | Single-Pole Model | Initial Double-Pole Model |
|---|---|---|
| Diameter of outer/inner rotor | 155.6/125.8 mm | |
| Diameter of outer/inner stator | 124.3/64 mm | |
| Stack length | 16 mm | |
| PM thickness/offset | 7/1 mm | |
| PM embrace | 0.82 | 0.91 |

**Table 2.** PM (permanent magnet) specification.

| Magnet Type | Ferrite |
|---|---|
| Remanence | 0.4689 T |
| Coercivity | −340,000 A/m |

### 2.2. Pole Separation Space

Figure 2 shows the schematics of the magnetizer structure for the double-pole PM. A high current applied to the coil results in the flux path shown in Figure 2. The magnetic flux generated by the current magnetizes the two poles in a magnet. Magnetic material is attached to the back of the magnet such that the magnet flux flows well; nonmagnetic material is attached to the front of the magnet to separate the two poles well. Therefore, the magnetic flux in the PM and the pole separation space is generated, as shown in Figure 2. Figure 3 displays the magnetization pattern of the single- and double-pole PMs, respectively, and their flux density distributions. The pole separation space includes a small flux. Although it is separated by nonmagnetic material, magnetic flux spreads in the pole separation space. In this study, we assume that the magnetic flux decreases linearly. As nonlinear modeling of this space is not only complex and difficult, but this space is also relatively less for the entire length of the magnet, the error ratio with the manufactured model is low [5]. Therefore, this space is calculated as twice the pole space of the single-pole model with a linear assumption and the magnitude of the magnetic flux is almost identical in the two models.

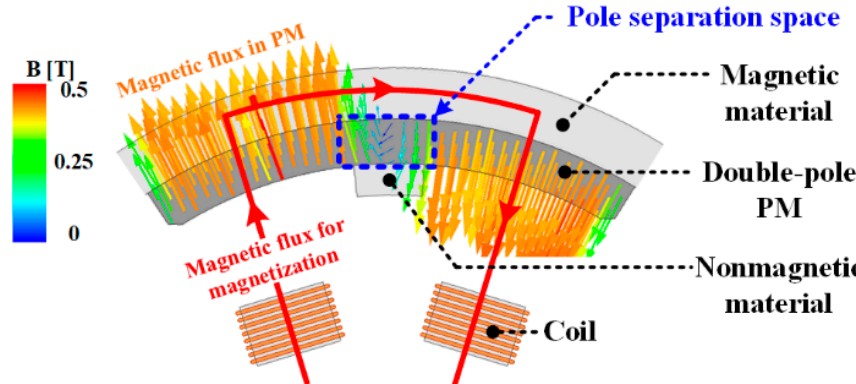

**Figure 2.** Schematics of the magnetizer structure for the double-pole PM.

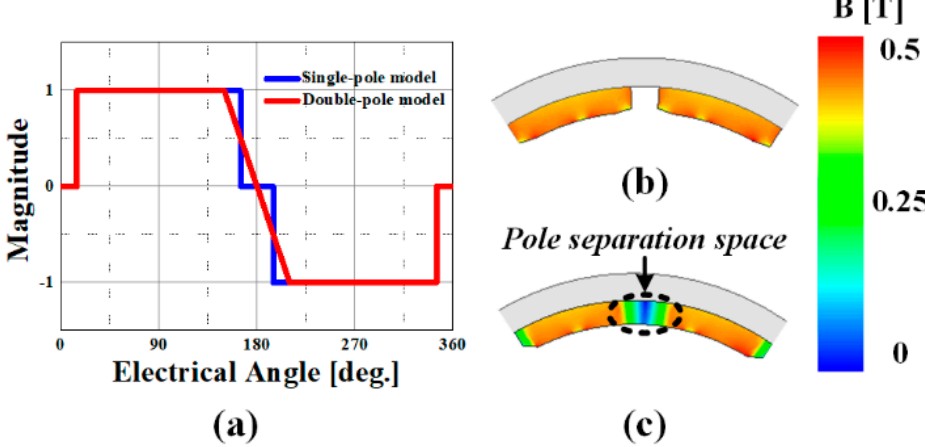

**Figure 3.** (**a**) Magnetization pattern function. Magnetic flux density distribution: (**b**) Single-pole model and (**c**) double-pole model.

*2.3. PM Offset*

The offset parts of the double-pole PM shown in Figure 4a are halved compared to the single-pole PM.

$$W_m = \frac{1}{2}B_m H_m V_{PM} \tag{1}$$

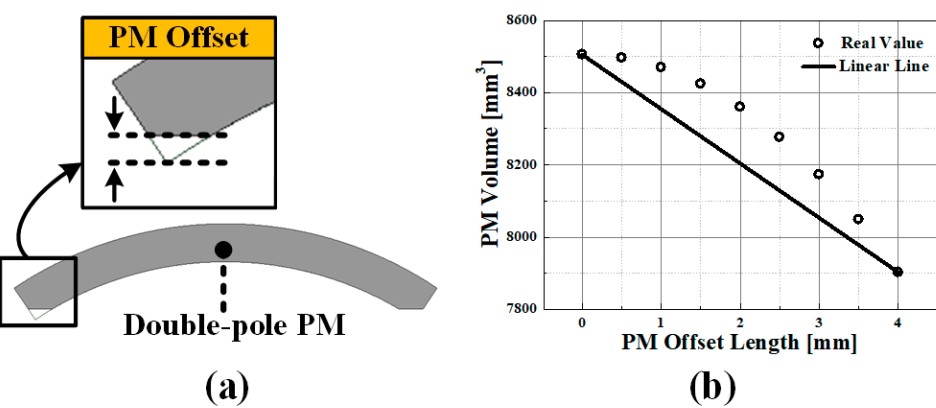

**Figure 4.** (**a**) PM offset and (**b**) relationship of the PM offset length and PM volume in the double-pole PM.

$W_m$ is the magnetic energy shown in Equation (1); $B_m$ and $H_m$ are the PM operating points of the magnetic flux density and magnetic field intensity, respectively; $V_{PM}$ is the volume of the PM [13]. From Equation (1), we can confirm that the volume of the PM is proportional to the magnetic energy. Therefore, the volumes of the single- and double-pole PM should be the same to have the same magnetic energy and the offset length of the double-pole model PM should be more than that of the single-pole PM. However, the volume of the double-pole PM is not linearly proportional to the offset length, as shown in Figure 4b. As it is difficult to arithmetically calculate proper offset length, the PM offset length of the initial double-pole model was selected to be the same value of the single-pole model. Figure 5 and Table 3 depict the analysis results of the single-pole and initial double-pole models, using FEM. Figure 5a,b show the results of the back EMF and cogging torque, respectively. In Figure 5 and Table 1, although the back EMF is almost identical and the error ratio is 1%, the cogging torque of the initial double-pole model is greater than that of the single-pole model and the error ratio of the cogging torque is 23%.

Thus, it cannot be stated that the two motors are the same in terms of the electromagnetic performance, and a design to reduce the cogging torque should be additionally performed.

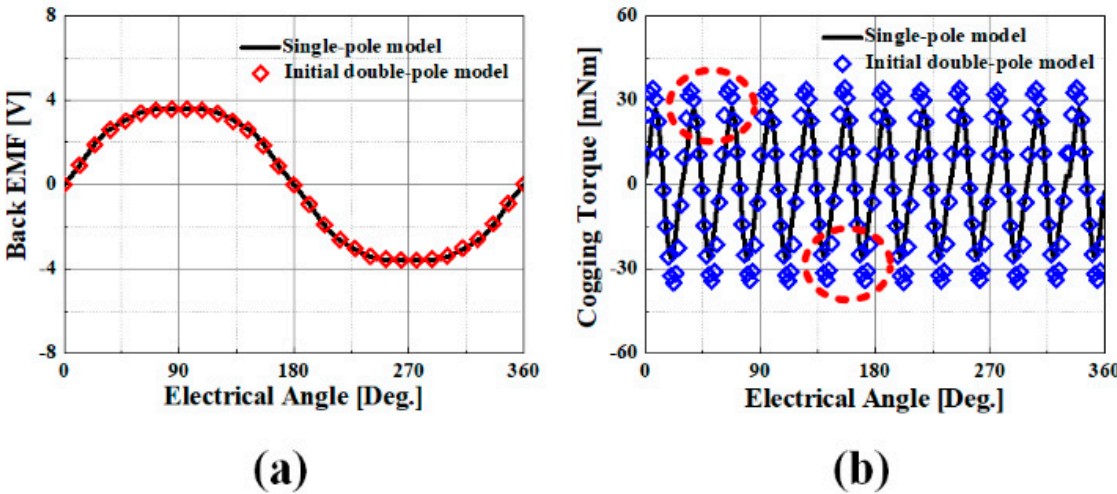

**Figure 5.** Analysis results of the single-pole and initial double-pole models: (**a**) Back EMF and (**b**) cogging torque.

**Table 3.** Analysis results of the single-pole and initial double-pole models.

| Item | Single-Pole Model | Initial Double-Pole Model |
|---|---|---|
| Back EMF | 2.79 V$_{rms}$ | 2.76 V$_{rms}$ |
| Cogging torque | 53 mNm | 69 mNm |

## 3. Optimal Design

RSM is generally used for setting the design variables to optimize the response values [14]. In this study, the objective of optimal design is to reduce the cogging torque while maintaining the back EMF. Cogging torque should be decreased to reduce the noise and vibration, and back EMF should be maintained for the electromagnetic performance. Therefore, in this optimal design, the response values are the back EMF and cogging torque, and the design variables are the previously selected pole separation space and PM offset.

### 3.1. CCD

CCD is the most common experimental design method in RSM. When there are two design variables, the number of the experimental points of the CCD is 15. These include a central point, four factorial points and four axial points. Table 4 lists all the values of the experimental points. $x_1$ and $x_2$ are the pole separation space and PM offset, respectively. The central point is generally the initial value of the design variables. However, because the initial value of $x_1$ 1 mm is much less, a value of 2 mm was re-selected. Factorial points are determined by the designer based on the initial values and the interval from the central point means $\pm 1$ level. Axial points are at $\pm \sqrt{2}$ level based on the central point and are used to estimate the curvature of the regression model [12].

$$y = \beta_0 + \beta_1 x_1 + \beta_2 x_1 + \beta_{11} x_1^2 + \beta_{22} x_2^2 + \beta_{12} x_1 x_2 \qquad (2)$$

The above equation is a general form of the regression model and is generally assumed as a quadratic equation. $y$ is the response value and $\beta_{ij}$ is the coefficient of each terms. $\beta_{ij}$ can be obtained from the response values of the experimental points through FEM. The final regression model is obtained by selecting the significant terms of $\beta_{ij}$ through variance analysis based on 90% reliability [12]. The coefficients of the selected terms are shown in Table 5. Using the regression model, we can obtain the contour plots of each response, as shown in Figure 6. The changes in each response, according to the design variables, can be observed.

**Table 4.** Experimental points.

| Point | Order | Pole Separation Space ($x_1$) | PM Offset ($x_2$) |
|---|---|---|---|
| Central point | 1 | 15 | 2 |
| Factorial point | 2 | 10 | 1 |
| | 3 | 20 | 1 |
| | 4 | 10 | 3 |
| | 5 | 20 | 3 |
| Axial point | 6 | 7.93 | 2 |
| | 7 | 22.07 | 2 |
| | 8 | 15 | 0.59 |
| | 9 | 15 | 3.41 |

**Table 5.** Coefficients of the regression model.

| Response Value | $\beta_0$ | $\beta_1$ | $\beta_2$ | $\beta_3$ |
|---|---|---|---|---|
| Back EMF ($y_1$) | 2.8372 | $7.853e^{-3}$ | $-0.03091$ | $-7.455e^{-4}$ |
| Cogging torque ($y_2$) | 312.562 | $-35.0899$ | $-2.76561$ | $1.11455$ |

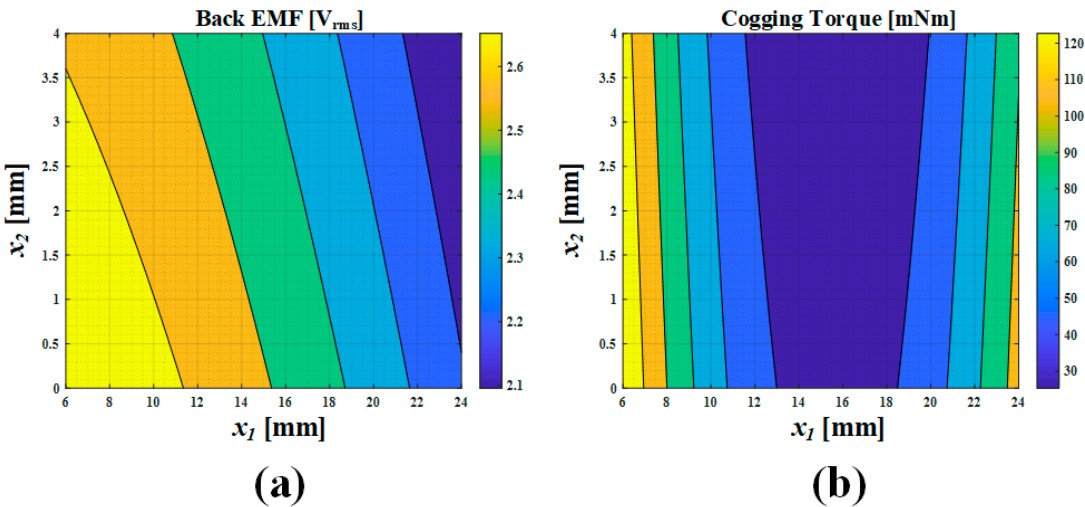

**Figure 6.** Contour plot: (**a**) Back EMF and (**b**) cogging torque.

*3.2. Multiple Response Optimal Method*

The multiple response optimal method, which is a type of RSM, can be applied for optimizing multiple response values simultaneously [15]. As previously mentioned, in this study, the objective of optimization is to maintain the back EMF $y_1$ and decreasing cogging torque $y_2$; the optimization settings are listed in Table 6. The target value of the back EMF was set to be the same as that of the single-pole model and the maximum and minimum values were less than 5% of the error ratio of the targeted value. Moreover, the maximum value of the cogging torque was set to be the same as that of the single-pole model, as shown in Table 6. The final optimal results are shown in Table 7. We can confirm that $x_1$ decreases whereas $x_2$ increases in the optimal process.

**Table 6.** Optimization settings.

| Item | Unit | Objective | Lower Limit | Target Value | Upper Limit |
|---|---|---|---|---|---|
| Back EMF ($y_1$) | V$_{rms}$ | Target value | 2.6 | 2.7 | 2.8 |
| Cogging torque ($y_2$) | mNm | Minimization | | 30 | 50 |

**Table 7.** Final optimal results.

| Design Variable | Initial Model | Optimal Model |
|---|---|---|
| Pole separation space ($x_1$) | 15 mm | 14.3 mm |
| PM offset ($x_2$) | 2 mm | 3.1 mm |

## 4. Experimental Verification and Discussion

### 4.1. Experimental Verification

Figure 7a displays the PMs and rotor core of the manufactured single-pole model. The single-pole PM has an overhang structure and the rotor core is integrated with housing. In this study, the models were analyzed by converting the 3D structures into 2D, considering the volume of the PM and rotor core [5,16,17]. Figure 7b shows the experimental setup for the back EMF measured using a drive motor at 1000 rpm. Figure 7c shows the experimental setup for the cogging torque measured using the torque sensor. Figure 8 presents the experimental and analysis results of the back EMF and cogging torque in the single-pole model, whereas Figure 9 presents the analysis result of the back EMF and cogging torque in the initial and optimal double-pole model. Table 8 lists the results of the electromagnetic performance of the single-pole model. Table 9 presents the results of the electromagnetic performances of the initial and optimal double-pole models. The line current and speed are measured and analyzed under the same terminal DC voltage and torque generation condition. We can confirm the FEM reliability from Figure 7 and Table 8. The error ratios of all the FEM parameters in the experiment are below 5%. Furthermore, we can confirm the optimization of the double-pole model from Figure 8 and Table 9. Although the back EMF and the other parameters are almost identical, the cogging torque is reduced. Finally, comparing the experimental results of the single-pole model with the analysis result of the double-pole model, the back EMF and the other parameters are almost identical, and their error ratios are below 3%. However, the cogging torque alone is lower than a wide error ratio. Thus, if the optimal double-pole model is manufactured, the cogging torque will be lower than that of the single-pole model, but the other electromagnetic performance parameters will be the same, as indicated by the high reliability of the FEM results and the results of optimal design.

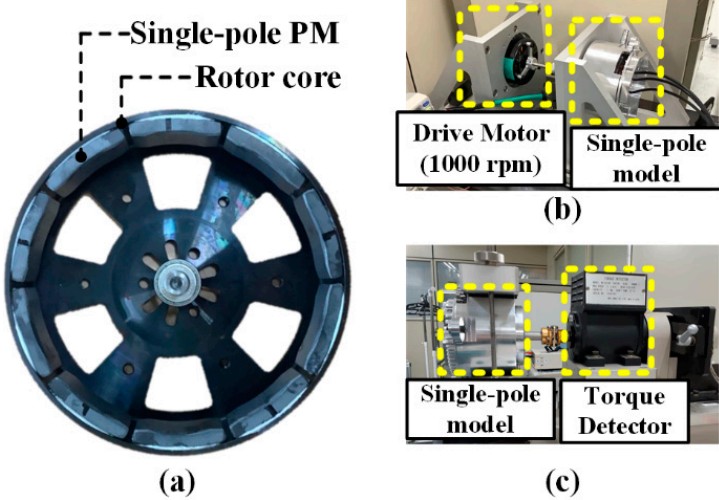

**Figure 7.** (**a**) PM and rotor core of the manufactured single-pole model. Experimental set up for: (**b**) Back EMF and (**c**) cogging torque.

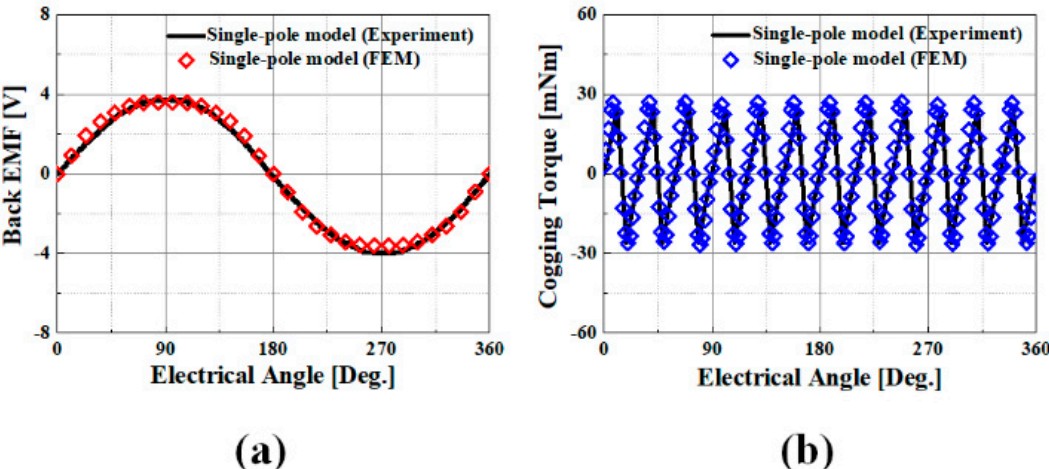

**Figure 8.** Experimental and analysis results of the single-pole model: (**a**) Back EMF and (**b**) cogging torque.

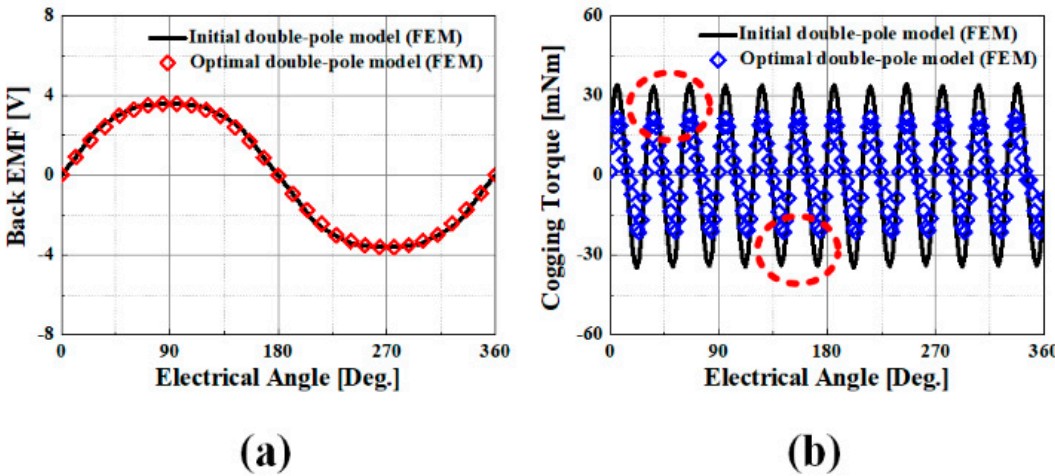

**Figure 9.** Analysis results of the initial and optimal double-pole models: (**a**) Back EMF and (**b**) cogging torque.

**Table 8.** Electromagnetic parameter results in the single-pole model.

| Parameters | Experiment | FEM (Error Ratio) |
|---|---|---|
| Back EMF [$V_{rms}$] | 2.77 | 2.79 (0.71%) |
| Cogging torque [mNm] | 50 | 53 (5.66%) |
| DC terminal voltage [V] | 13.5 | 13.5 |
| Line current [$A_{rms}$] | 58.17 | 58.96 (1.34%) |
| Torque [Nm] | 2.38 | 2.42 (1.65%) |
| Speed [rpm] | 2594 | 2670 (2.85%) |
| Efficiency [%] | 90.91 | 92.28 (1.48%) |

**Table 9.** Electromagnetic parameter analysis results in the double-pole model.

| Parameters | Initial Model (Error Ratio) | Optimal Model (Error Ratio) |
|---|---|---|
| Back EMF [$V_{rms}$] | 2.76 (0.36%) | 2.78 (0.36%) |
| Cogging torque [mNm] | 69 (27.54%) | 44 (13.54%) |
| DC terminal voltage [V] | 13.5 | 13.5 |
| Line current [$A_{rms}$] | 59.15 (1.66%) | 58.96 (1.34%) |
| Torque [Nm] | 2.43 (2.06%) | 2.43 (2.06%) |
| Speed [rpm] | 2679 (3.17%) | 2678 (3.14%) |
| Efficiency [%] | 92.18 (1.37%) | 92.19 (1.38%) |

*4.2. Discussion*

In this section, the analysis of the back EMF and cogging torque results is discussed.

$$e_{abc} = k_e \phi_{abc} \omega_m \tag{3}$$

$e_{abc}$ is the phase back EMF, as shown in Equation (3). In Equation (3), $k_e$ is the coefficient of the back EMF, $\phi_{abcf}$ is the phase flux by PM and $\omega_m$ is the angular speed [8]. When $\omega_m$ is constant at 1000 rpm in all the models, $e_{abc}$ is proportional to $\phi_{abcf}$ in Equation (3). $x_1$ and $x_2$ are inversely proportional to $\phi_{abcf}$. Although $x_1$ of the optimal double-pole model is less than that of the single-pole model and initial double-pole model, $x_2$ is greater. Figure 7a displays the PMs and rotor cores of the manufactured single-pole model. The single-pole PM has an overhang structure and the rotor core is integrated with housing. Therefore, $x_2$ and the back EMF of the optimal double-pole model are almost identical to those of the single-pole and initial double-pole models. $T_{cog}$ is the cogging torque, as shown in Equation (4), and $\Re$ is the reluctance [18].

$$T_{cog} = \frac{1}{2} \phi_{abcf}^2 \frac{d\Re}{d\theta} \tag{4}$$

As mentioned above, $\phi_{abcf}$ is almost identical in the single-pole model and in the initial and optimal double-pole model. However, as $x_2$ is inversely proportional to $\frac{d\Re}{d\theta}$ [18] and $x_2$ in the optimal double-pole model is greater, the cogging torque of the optimal double-pole model is less than that of the manufactured single-pole model and the initial model.

## 5. Conclusions

This study presented an optimal design for a double-pole magnetization BLDC motor. An initial double-pole model was selected based on the single-pole model's PM. The pole separation space was selected as twice of the single-pole model's pole space due to linear decreasing assumption of flux in the pole separation space. The same PM offset was selected because it was difficult to arithmetically calculate the correct offset length due to the nonlinear relationship between the PM volume and offset. The analysis results of the single-pole and initial double-pole models indicated that the back EMF was almost the same but the cogging torque of the initial double-pole model was greater. Therefore, a suitable pole separation space and PM offset were selected as per the optimal design; the multiple response optimal method was used in this study. The objective of the optimal design was to maintain the back EMF and decrease the cogging torque. The optimization results were verified by comparing the experimental and analysis results of the single-pole model with the analysis results of the optimal double-pole model. Additionally, the electromagnetic performance was also maintained. Consequently, the double-pole model can provide almost same or better electromagnetic performance compared with the single-pole model using the optimal method. Moreover, the double-pole model has driving stability and manufacturing advantages compared to the single-pole model. Therefore, the double-pole magnetization BLDC motor can sufficiently replace the single-pole magnetization BLDC motor which has many poles.

**Author Contributions:** J.-Y.C.: conceptualization, review and editing; H.-S.S.: analysis and original draft preparation; G.-H.J.: experiment and conceptualization; K.-H.J.: analysis method review; S.-K.C.: analysis method review; H.-J.S.: experiment. All authors have read and agreed to the published version of the manuscript.

**Funding:** This work was supported by the National Research Foundation of Korea (NRF) grant funded by the Korea government (MSIT). (No. 2020R1A2C1007353), and by Korea Electric Power Corporation. (Grant number: R20XO02-38).

**Institutional Review Board Statement:** Not applicable.

**Informed Consent Statement:** Not applicable.

**Data Availability Statement:** The data presented in this study are available on request from the corresponding author.

**Conflicts of Interest:** The authors declare no conflict of interest.

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
