# Peer review of "Optimal Design of Double-Pole Magnetization BLDC Motor and Comparison with Single-Pole Magnetization BLDC Motor in Terms of Electromagnetic Performance"

_machines, doi:10.3390/machines9010018_

Round 1

Reviewer 1 Report

Dear authors,

Here are some questions and comments:

Please, enlarge the introduction. Were these types of motor or similar ones optimized previously?

Please, designate the winding phases and their direction in the slots in Fig.1.

Please provide the main motor’s dimensions such as outer radius, stack length, the thickness of the motor.

Which type of magnets is used in the motor? Which are their remanence and coercitivity?

Does Fig. 2 present real ratios between the magnetizer dimensions?

Please provide the figure providing the magnetic flux distribution in the magnets during their magnetization.

Provide some information on the mathematical model used to simulate the motors.

Add units in table 4.

Reviewer 2 Report

The paper is very interesting and well prepared, these changes must be made to improve quality:

1 .. More state of the art is missing to improve the bibliographic review in the introduction, where the bases of this research are shown.
2 .. The problem statement is not clear.
3 .. The objectives of the investigation are not clear.
4 .. the mathematical supports are poor.
5 .. The letters (a), (b) of all the figures have different proportions.
6 .. The labels in figures 4, 5, 8, 9 are very large font.
7. The titles in Figure 6 are in very large type.
8 .. You should put more results.
9 .. Finely improve the conclusions.

Reviewer 3 Report

The article is well prepared and written. The drawings are legible. The results are well presented. I have no comments on this version of the article

Author Response

Thanks for your comments. I really appreciate it.

Round 2

Reviewer 1 Report

Dear authors,

The paper has been significantly improved and can be published.

Author Response

(The authors gave the same response as above.)

Reviewer 2 Report

Congratulations on the excellent research. And thank you very much for the corrections made in the first round.

Finally, to improve the quality of the article, the formulation of the problem supported with more updated bibliographic references should be further improved. You should have many more references after 2015.

Best regards
